# Microglia Are Necessary to Regulate Sleep after an Immune Challenge

**DOI:** 10.3390/biology11081241

**Published:** 2022-08-19

**Authors:** Rachel K. Rowe, Tabitha R. F. Green, Katherine R. Giordano, J. Bryce Ortiz, Sean M. Murphy, Mark R. Opp

**Affiliations:** 1Department of Integrative Physiology, University of Colorado, Boulder, CO 80301, USA; 2Barrow Neurological Institute at Phoenix Children’s Hospital, Phoenix, AZ 85016, USA; 3Department of Child Health, University of Arizona College of Medicine-Phoenix, Phoenix, AZ 85004, USA; 4Phoenix Veterans Affairs Health Care System, Phoenix, AZ 85012, USA

**Keywords:** glia, inflammation, sleep–wake disturbances, microglia depletion, plx5622

## Abstract

**Simple Summary:**

Microglia are cells in the brain that mediate inflammation, but little is known about the role of microglia in sleep following an inflammatory challenge. We hypothesized that mice with depleted microglia would sleep less following an injection of a particle that causes inflammation in the body and brain. Brains were collected, the shape and complexity of microglia were measured, and biological data were collected to determine how long the mice slept. Mice fed PLX5622 (PLX), a compound that depletes microglia, had an immediate increase in sleep that resolved by week 2. After the first injection to cause inflammation, mice with depleted microglia slept more than control mice that had normal microglia. The PLX diet was switched to a standard mouse diet and microglia returned to the brain. While microglia were returning, there were no changes in sleep. However, following a second injection to cause inflammation, mice that had new microglia slept the same as control mice during the dark period, but had longer episodes of sleep during the light period. Comparing sleep after the first and second inflammation injection, control mice had different sleep patterns but no change in total time spent sleeping. Mice with new microglia slept less during the dark period. New microglia also had a different shape than microglia in control mice. We conclude that microglia are necessary to regulate sleep after an inflammatory challenge.

**Abstract:**

Microglia play a critical role in the neuroimmune response, but little is known about the role of microglia in sleep following an inflammatory trigger. Nevertheless, decades of research have been predicated on the assumption that an inflammatory trigger increases sleep through microglial activation. We hypothesized that mice (*n* = 30) with depleted microglia using PLX5622 (PLX) would sleep less following the administration of lipopolysaccharide (LPS) to induce inflammation. Brains were collected and microglial morphology was assessed using quantitative skeletal analyses and physiological parameters were recorded using non-invasive piezoelectric cages. Mice fed PLX diet had a transient increase in sleep that dissipated by week 2. Subsequently, following a first LPS injection (0.4 mg/kg), mice with depleted microglia slept more than mice on the control diet. All mice were returned to normal rodent chow to repopulate microglia in the PLX group (10 days). Nominal differences in sleep existed during the microglia repopulation period. However, following a second LPS injection, mice with repopulated microglia slept similarly to control mice during the dark period but with longer bouts during the light period. Comparing sleep after the first LPS injection to sleep after the second LPS injection, controls exhibited temporal changes in sleep patterns but no change in cumulative minutes slept, whereas cumulative sleep in mice with repopulated microglia decreased during the dark period across all days. Repopulated microglia had a reactive morphology. We conclude that microglia are necessary to regulate sleep after an immune challenge.

## 1. Introduction

Although sleep research, historically, has been neuron centric, contemporary evidence indicates that glia actively participate in sleep regulation [1,2]. Glia, specifically microglia, may facilitate the bi-directional interactions between neuroinflammation and sleep. Multiple studies have demonstrated a critical role for microglia in the neuroimmune response [3,4,5,6,7]; yet, little is known about their contribution to inflammation-mediated alterations in sleep. Consequently, a substantial knowledge gap exists regarding the molecular mechanisms by which an inflammatory challenge alters sleep.

Inflammatory triggers, such as infection, initiate an immune response that results in microglia reactivity. Microglia are a major cellular component in the innate immune system in the central nervous system (CNS), actively respond to injury, infection, or disease, and play a critical role in neuroinflammation [8,9,10]. Following an inflammatory trigger, microglia can rapidly respond to induce a protective immune response. This response consists of, among other things, a transient upregulation of inflammatory molecules and neurotrophic factors, clearance of cellular debris, and the initiation of restorative processes that promote homeostasis [8,11,12]. In this manner, microglia become reactive, protect CNS functions, and remove damaged cells from sites of acute insult. However, prolonged reactivity of microglia can result in neurotoxicity and tissue damage [13,14,15]. Thus, inflammation is a necessary and beneficial response to a stimulus, but when unregulated, it can lead to deleterious effects on health and recovery.

The protracted reactivity of microglia following an inflammatory challenge can cause high levels of inflammatory cytokines, which can prolong the inflammatory response and worsen functional outcomes [16]. Cytokines not only communicate with other immune cells but are also key messengers in immune–brain interactions, including sleep [4,5,17,18]. Extensive prior research demonstrates that many pro-inflammatory cytokines, such as interleukin-6 (IL6), IL1β, and tumor necrosis factor-α (TNFα), are also pro-somnogenic [3,4]. Conversely, anti-inflammatory cytokines (IL4, IL10, IL13, the IL1 receptor antagonist) are anti-somnogenic (reviewed [19,20]).

Sleep disturbances in healthy adults induce inflammation [21,22,23,24,25,26,27,28,29,30,31], characterized, in part, by increased production of IL1β, IL6, and TNFα. This inflammatory response is likely mediated by reactive microglia. Microglia are both targets and sources of sleep-altering cytokine signaling [32] and an established link exists between disturbed sleep and microglial reactivity [33,34,35,36,37]. Microglia play a role in the brain’s response to disturbed sleep and sleep disorders (reviewed in [1,2,36,38]). For instance, five days of sleep restriction activates microglia in rat hippocampus [33]. Chronic sleep restriction de-ramifies microglia [34] and increases the density of microglia in brain regions involved in sleep/wake regulation [35]. Minocycline, a compound that attenuates microglial reactivity, suppresses rebound sleep in mice after sleep deprivation [39]. The collective results of those studies suggest that microglia may play a role in the neuroimmune response to sleep.

This study targets the enduring knowledge gap by investigating the functional role of microglia in sleep after an immune challenge. Lipopolysaccharide (LPS) induces inflammation, propagates cytokine release by reactive microglia, and causes behavioral changes and alterations in sleep [40,41,42]. To further elucidate a role for microglia as mediators of sleep responses to LPS challenge, we used PLX5622 (PLX) to deplete microglia. PLX inhibits the colony-stimulating factor-1 receptor (CSF-1R), which is critical for microglia survival. We determined changes in sleep during the period of microglia depletion and then administered LPS as an immune challenge. We hypothesized that mice with depleted microglia would sleep less following an intraperitoneal administration of LPS. We then withdrew PLX to allow microglia to repopulate and determined sleep after a second LPS injection. We further hypothesized that the sleep response to repeated exposure of LPS in mice with repopulated microglia would be comparable to the sleep response of control mice in which microglia had not been manipulated.

## 2. Materials and Methods

### 2.1. Rigor

All animal studies were conducted in accordance with the guidelines established by the internal Institutional Animal Care and Use Committee (IACUC) at the University of Arizona and the National Institutes of Health (NIH) guidelines for the care and use of laboratory animals. Studies are reported following Animal Research: Reporting In Vivo Experiments (ARRIVE) guidelines [43]. Animals were randomly assigned to manipulation groups before the initiation of the study to ensure equal distribution across groups. Sleep data collection stopped at pre-determined final endpoints based on days post-LPS injection for each animal. Determination of sleep–wake behavior based on physiological parameters and the quantification of microglial morphologies were performed by investigators blind to the experimental treatments.

### 2.2. Animals

Adult (8–10-week-old) male C57BL/6J mice (20–25 g, Jackson Laboratories, Bar Harbor, ME, USA) were used for all experiments (*n* = 30). Mice were singly housed and were maintained on a 14 h light (200 lux, cool white fluorescent light; zeitgeber time [ZT] 0): 10 h dark (ZT 14) cycle at an ambient temperature of 23 °C ± 2 °C.

### 2.3. Study Design, Plexxikon Administration, and LPS Injections

The study design is summarized in Figure 1. Mice were acclimated to non-invasive piezoelectric sleep cages and fed a normal diet of standard rodent chow for 7 days. Following the acclimation period, mice were randomly assigned to control diet (AIN-76A) or PLX5622 (PLX) diet (1200 mg/kg) formulated in AIN-76A rodent chow for a 21-day microglia depletion period. Microglia depletion was confirmed in the PLX group (Appendix A). Mice continued to receive control or PLX diet and were subjected to an inflammatory challenge using lipopolysaccharide (LPS; *E. coli* 0111:B4, Sigma-Aldrich, St. Louis, MO, USA). LPS was made as a stock solution in sterile saline (0.9%) and injected intraperitoneally (i.p.) at 0.4 mg/kg in a volume of 0.05 mL between ZT3 and ZT4 (first LPS administration; LPS1). At 4 days post-LPS injection, all mice were switched back to a normal diet to allow repopulation of microglia in the PLX group. Following a 10-day repopulation period, mice were subjected to a second LPS injection (LPS2) and all mice remained on normal diet until tissue was collected 7 days after the second LPS injection. We did not include a vehicle control for the LPS injections because our comparison of interest was PLX-LPS vs. control-LPS. However, our within-subjects protocol allowed us to determine changes after LPS administration relative to baseline values for each animal during the depletion and repopulation period. Throughout the study, access to food and water remained ad libitum. Following LPS1, 1 mouse died and 2 mice on PLX were euthanized for excessive weight loss and dehydration. Following LPS2, 1 mouse died. There was a technical error in tissue collection that resulted in one brain not being processed for immunohistochemistry and microglial analyses. Final sample sizes for each experiment are included in the figure legends.

### 2.4. Collection of Physiological Sleep–Wake Parameters

Sleep and wake physiological parameters were collected using a non-invasive piezoelectric sleep cage system (Signal Solutions, Lexington, KY, USA), which classified sleep behavior according to previously described methods [44,45,46]. This non-invasive method has been validated with electroencephalogram (EEG) and human observations and has demonstrated a classification accuracy (sleep vs. wake) of >90% in mouse sleep research [47]. Each cage has an open bottom that allows the mouse to be placed directly on a Polyvinylidine Difluoride sensor on the cage floor. These sensors are coupled to an input differential amplifier to generate pressure signals. Regular breathing movements characterize sleep (3 hertz, regular amplitude signals; [48]), whereas signals from awake mice are of higher amplitude with irregular spiking associated with volitional movements. In this study, the piezoelectric signals were analyzed over tapered eight-second windows at a two-second interval, from which a decision statistic was computed and classified by a linear discriminate classifier as “sleep” or “wake”. Data collected from the sleep cages were binned at each hour using a rolling average of the percentage sleep. Data were also binned by length of individual sleep bouts to calculate the hourly mean bout length (duration in seconds). To be considered a bout, a minimum of two consecutive epochs had to be scored as sleep. Cumulative minutes slept within each experimental period (e.g., week 1, day 1) were also calculated.

### 2.5. Tissue Collection and Immunohistochemistry

At a pre-determined time point (7 days following LPS2), brain tissue was collected. Mice were injected intraperitoneally with Euthasol (0.002 mL/g, Patterson Veterinary, Greeley, CO, USA) and perfused with iced 1× phosphate-buffered saline (PBS). Brains were removed and post fixed in 4% paraformaldehyde (PFA) for 24 h. Next, brains were cryoprotected in successive concentrations of sucrose (15%, 30%). Hemispheres were submerged in a mold filled with optimal cutting temperature compound using the Megabrain technique [49]. Brains were flash frozen at −48 °C in isopentane. Fixed brains were cryosectioned in the coronal plane at 40 µm and were mounted on slides and stored at −80 °C. Slides were defrosted and placed in the oven at 56 °C for 3 h before staining. Slides were washed in PBS and incubated in blocking solution (4% Normal horse serum [NHS], 0.1% Triton-100 in PBS) for 60 min. Slides were then incubated in primary antibody solution (rabbit anti-Iba1; WAKO cat #019919741 at 1:1000 concentration in 1% NHS, 0.1% triton-100 in PBS) overnight at 4 °C. Slides were washed in PBS and 1% tween and incubated in secondary antibody solution (biotinylated horse anti-rabbit IgG (H+L); vector BA-1100 at 1:250 concentration in 4% NHS and 0.4% triton-100 in PBS) for 60 min at room temperature. Slides were washed in PBS and incubated in hydrogen peroxide for 30 min and then incubated in ABC solution (Vectastain ABC kit PK-6100) for 30 min. 3,3′-Diaminobenzidine (Vector DAB peroxidase substrate kit SK-4100) was applied for 10 min. Slides were then placed in tap water followed by successive concentrations of alcohol for 5 min each (70%, 90%, 100%), incubated in Citrosolve, and cover slips were applied using dibutyl phthalate polystyrene xylene (DPX) mounting medium.

### 2.6. Imaging and Microglia Skeletal Analysis

Z-stacked photomicrographs in pre-determined regions of interest were taken using a 40× objective lens on a Zeiss Imager A2 microscope via AxioCam MRc5 digital camera and Neurolucida 360 software (MicroBrightfield, Williston, VT, USA), with consistent brightness, exposure, and Z-stack height. Three randomly selected brain slices located between bregma and lambda from the left hemisphere of each animal were imaged. For each animal, three cortical regions of interest were analyzed per brain slice. Regions of interest included retrosplenial, somatosensory, and entorhinal cortices. We selected three cortical regions because it is well documented that LPS administration induces microglia reactivity in the cortex [50]. Furthermore, alterations in sleep lead to microglia reactivity in the mouse cortex [34]. Iba1 staining was analyzed using the ImageJ skeletal analysis plugin and a previously published protocol for microglia analyses (Morrison et al., 2017; Young & Morrison, 2018). Photomicrographs were converted to 8-bit and an FFT bandpass filter was applied. The brightness and contrast were adjusted and the unsharp mask and despeckle functions were applied. The threshold was then adjusted to create a binary image, followed by the despeckle, close, and remove outliers functions. The photomicrograph was then skeletonized and the microglial branch length, number of microglial processes, and number of process endpoints were calculated as an average per cell across the entire field of view. Microglial cell counts were performed manually by an investigator blinded to the experimental groups.

### 2.7. Statistical Analyses

To investigate sleep differences between treatment groups across time points within each experimental phase (depletion, LPS1, repopulation, and LPS2) and microglial differences between treatment groups at the terminal time point, we fit hierarchical generalized linear mixed models [51,52]. We fit all models in the frequentist framework using the package glmmTMB in the R statistical computing environment [53,54]. The outcome measures of interest were either percentages or overdispersed counts, so we specified Beta or negative-binomial error distributions, respectively, in the corresponding models [55,56].

Percentage sleep and bout lengths were sequential time-series data with cyclical light–dark period trends within which observations close to each other in time are expected to be more similar than observations further apart in time (i.e., dependency among observations) [57,58]. Additionally, the rodent sleep cycle is expected to have temporally varying nonlinear interactions with predictor variables and temporally varying nonlinear effects on outcome measures [46,59]. Therefore, we included a six-knot basis spline for time (ZT) in all percentage sleep and bout length models to accommodate the expected similarity among observations within a period and the nonlinear effects of time [60,61].

For both percentage sleep and bout lengths, we subdivided the data by period (i.e., light and dark periods) and fit period-specific models with a three-way interaction among the categorical variables treatment, days (LPS1 and LPS2) or weeks (depletion and repopulation), and the continuous variable ZT with a basis spline [46,59]. For cumulative sleep, we also subdivided the data by period and fit period-specific models, but we instead included a two-way interaction between treatment and days or weeks, disregarding ZT. In all sleep outcome models, we included random intercepts for individual mice nested within cohorts to account for the dependency of observations from the same individual in a given cohort [51,52]. For all three microglial morphology outcome measures, we fit models that included only a treatment main effect with random intercepts for individual mice crossed with random intercepts for the binned number of cells that were used to derive microglial outcome metrics for each animal.

We based inferences on a combination of coefficient estimates (β), differences between estimated marginal means (∆), Cohen’s *d* effect sizes [62], and *p*-values following Tukey’s adjustments for multiple comparisons [63], all of which were obtained using the package emmeans in R [64]. Additionally, to quantify temporal pattern differences within each period for percentage sleep and sleep bout lengths, we estimated nonparametric Spearman’s rank correlation (ρ) using the package stats in R [53], where −0.60 < ρ < 0.60 and *p* > 0.05 denoted statistically significant pattern discrepancies (i.e., −0.60 > ρ > 0.60 and *p* < 0.05 represent high correlation that reflect pattern similarity).

## 3. Results

### 3.1. Microglia Depletion Resulted in a Transient Sleep Increase during the Light Period

Mice on the PLX diet slept more during week 1 compared to mice on the control diet (Figure 2A–F; see Appendix A for corresponding differences between estimated marginal means, effect sizes, Spearman’s rank correlation, and *p*-values). During the light period, mice on PLX had a higher percentage sleep (Figure 2A), longer bouts (Figure 2C), and more cumulative minutes slept (Figure 2E) compared to mice on the control diet. There were no differences in means or temporal pattern changes of sleep during the dark period in week 1 (Figure 2B,D,F) or the light or dark period for weeks 2 or 3 (Figure 2A–F).

### 3.2. Mice with Depleted Microglia Slept More during the Dark Period Following LPS1 Administration

Mice on the PLX diet slept more following LPS1 compared to mice on the control diet (Figure 3A–F; see Appendix A for corresponding differences between estimated marginal means, effect sizes, Spearman’s rank correlation, and *p*-values). On day 1 post-LPS1, Mice on the PLX diet had temporal pattern changes of percent sleep during both the light and dark period (Figure 3A,B), temporal pattern changes in bout length during the dark period (Figure 3D), and an overall increase in percent sleep, bout length, and cumulative sleep during the dark period (Figure 3B,D,F).

On day 2 post-LPS1, there were no differences in means or temporal pattern changes in percent sleep, bout length, or cumulative sleep during the light period (Figure 3A,C,E). During the dark period, mice on PLX had increased percent sleep (Figure 3B) and more cumulative minutes slept compared to mice on the control diet (Figure 3F).

On day 3 post-LPS1, mice on PLX had temporal pattern changes in percent sleep (Figure 3A) and bout length, with longer bouts during the light period (Figure 3C) and more cumulative sleep during the dark period (Figure 3F). There were no differences in means for percent sleep and cumulative sleep during the light period (Figure 3A,E) or percent sleep and bout length during the dark period (Figure 3B,D).

On day 4 post-LPS1, mice on PLX had temporal pattern changes in bout length during the light period (Figure 3C) and longer bouts and more cumulative sleep during the dark period (Figure 3D,F). There were no temporal pattern changes in percent sleep or differences in the means of percent sleep, bout length, or cumulative sleep during the light period (Figure 3A,C,E). There were no temporal pattern changes in percent sleep or bout length and no differences in the means of percent sleep during the dark period (Figure 3B,D,F).

### 3.3. Sleep Changes during Microglia Repopulation Were Nominal

During the repopulation period, sleep was comparable between mice that were on the PLX diet and mice that had been on the control diet (Figure 4A–E; see Appendix A for corresponding differences between estimated marginal means, effect sizes, Spearman’s rank correlation, and *p*-values). In week 1 of the repopulation period, mice that were on the PLX diet had longer bouts during the light period (Figure 4C). There were no other temporal pattern changes or differences in the means of percent sleep, bout length, or cumulative sleep in the light or dark period for week 1 or week 2 (Figure 4A–E).

### 3.4. In Mice with Repopulated Microglia, LPS2 Resulted in Longer Bouts during the Light Period and No Differences in Sleep during the Dark Period Compared to Controls

Following repopulation of microglia, mice that were previously on the PLX diet had longer bouts during the light period, but similar sleep during the dark period across all time points compared to mice that were on the control diet (Figure 5A–F; see Appendix A for corresponding differences between estimated marginal means, effect sizes, Spearman’s rank correlation, and *p*-values). On days 2 and 4 post-LPS2, mice previously on the PLX diet had temporal pattern changes to percent sleep during the light period compared to control mice (Figure 5A). On days 2, 3, 4, and 5 post-LPS2, mice previously on the PLX diet had longer bout lengths during the light period compared to control mice (Figure 4C). On day 4 post-injection, mice previously on the PLX diet also had temporal pattern changes to bout lengths during the light period (Figure 5C). There were no differences in the means of percent sleep during the light (Figure 5A) or dark periods (Figure 5B). There were no differences in the bout length or temporal pattern changes in bout lengths on days 1 and 7 during the light period (Figure 5C) or any day during the dark period (Figure 5D). There were no differences in cumulative sleep across all time points post injection during the light (Figure 5E) or dark periods (Figure 5F).

### 3.5. In Control Mice, LPS2 Did Not Change the Total Time Spent Sleeping but Resulted in Temporal Pattern Changes in Percent Sleep and Bout Lengths

LPS2 altered the sleep pattern, primarily during the light period, in mice that had previously been on the control diet (Figure 6A–F; see Appendix A for corresponding differences between estimated marginal means, effect sizes, Spearman’s rank correlation, and *p*-values). On day 1, mice had longer bouts during the light period after LPS2 compared to LPS1, with temporal pattern changes in bout lengths during both the light and dark periods (Figure 6C,D). On day 2, mice had temporal pattern changes in percent sleep and bout lengths during the light period and longer bouts after LPS2 compared to LPS1 (Figure 6C,D).

On days 3 and 4, LPS2 resulted in temporal pattern changes to percent sleep and bout length during the light period compared to LPS1 (Figure 6C,D). There were no temporal changes to percent sleep on days 2, 3, and 4 during the dark period (Figure 6B). There were no differences in means of percent sleep or bout length and no temporal changes to bout length during the dark period at any day post LPS2 (Figure 6B,D). There were no differences in cumulative sleep during the light or dark periods at any day post-LPS2 (Figure 6E,F).

### 3.6. In Mice with Repopulated Microglia, LPS2 Resulted in Longer Bouts during the Light Period and Less Sleep during the Dark Period

Following the repopulation of microglia in mice that received the PLX diet, LPS2 resulted in significantly less sleep during the dark period compared to sleep following LPS1 (Figure 7A–F; see Appendix A for corresponding differences between estimated marginal means, effect sizes, Spearman’s rank correlation, and *p*-values). On day 1, mice with repopulated microglia had longer bouts and temporal pattern changes to percent sleep and bout length during the light period following LPS2 compared to LPS1 (Figure 7A,C). During the dark period, mice had a lower percent sleep with temporal pattern changes in bout length and less cumulative minutes slept following LPS2 compared to LPS1 (Figure 7B,D,F).

On day 2, mice with repopulated microglia had longer bouts with temporal changes in bout length but no differences in percent sleep or cumulative sleep during the light period following LPS2 compared LPS1 (Figure 7A,C,E). During the dark period, mice had a lower percent sleep with fewer cumulative minutes slept, but no differences in bout lengths, following LPS2, compared to LPS1 (Figure 7B,D,F).

On day 3, mice with repopulated microglia had longer bouts but no temporal changes in percent sleep or bout length and no differences in the means of percent sleep or cumulative minutes slept during the light period following LPS2 compared to LPS1 (Figure 7A,C,E). During the dark period, mice had a lower percent sleep, longer bout length, and fewer cumulative minutes slept, but had no temporal pattern changes in percent sleep or bout length following LPS2 compared to LPS1 (Figure 7B,D,F).

On day 4, mice with repopulated microglia had longer bouts but no differences in the means of percent sleep and cumulative minutes slept or temporal pattern changes in percent sleep or bout length during the light period following LPS2 compared to LPS1 (Figure 7A,C,E). During the dark period, mice had a lower percent sleep and slept fewer cumulative minutes but had no temporal pattern changes in percent sleep and no differences in bout lengths following LPS2 compared to LPS1 (Figure 7B,D,F).

### 3.7. Repopulated Microglia Had a Reactive Morphology with Fewer Endpoints per Microglial Cell Compared to Control Mice

Microglial morphology was assessed on brain tissue stained with Iba1 (Figure 8A,B). There were no differences in the mean number of microglia per field of view in mice with repopulated microglia compared to controls (Figure 8C; see Appendix A for corresponding differences between estimated marginal means, effect sizes, and *p*-values). Mice with repopulated microglia had 140.85 ± 9.81 microglia cells/mm^2^ compared to controls that had 139.12 ± 11.78 microglia cells/mm^2^. There were no differences in the mean process length per microglial cell (Figure 8D) or mean microglial branch length (Figure 8E) in mice with repopulated microglia compared to controls. There were fewer endpoints per microglial cell in mice with repopulated microglia compared to controls (Figure 8F).

## 4. Discussion

Inflammation is elevated in individuals with sleep disturbances, such as insomnia [65,66], obstructive sleep apnea [67,68,69], and restless legs syndrome [70,71]. Furthermore, inflammatory diseases, including cancer [72,73,74,75], stroke [76,77], traumatic brain injury [78,79,80,81], Alzheimer’s disease [82,83,84,85,86], and autoimmune disorders [87,88,89,90], increase the risk for sleep disturbances. Therefore, a bi-directional relationship exists between sleep and inflammation that can negatively impact health. Decades of research have been predicated on the assumption that an inflammatory trigger increases sleep through microglial activation. Microglia are quick to respond to an immunological challenge in the brain and rapidly respond to infection or injury [50,91]. Although many studies demonstrate a critical role for microglia in neuroimmune processes [3,4,5,6,7], few studies have investigated the role of microglia in sleep under inflammatory conditions. Identifying a mechanism to regulate inflammation-induced sleep is a plausible target to improve health and quality of life.

Microglia also have non-immunological functions and play a vital role in maintaining homeostasis in the healthy brain [91], but their role in sleep under non-inflammatory conditions is understudied. During the first week of microglia depletion, we found that mice lacking microglia slept more, with longer bout durations, compared to mice on the control diet. This increase in sleep resolved and there were no differences in sleep between mice on PLX or the control diet in weeks 2 or 3 of the depletion period. PLX administration rapidly depletes microglia with significant reductions in microglia (95%) achieved within 3 days of administration, at a dose of 1200 mg/kg [92], the same dose used in our study. Therefore, most microglia were eliminated during week 1 of the depletion period in our study. The mechanism of eliminating and clearing microglial cells from the brain following PLX-induced depletion is not fully characterized, but there is evidence to support that microglia are eliminated as a result of apoptosis [93]. We postulate that during week 1 of the depletion period, microglia were rapidly eliminated and the wide-spread cell death and clearance of waste transiently altered sleep. Inflammatory cytokines play a role in phagocytosis and clearance of apoptotic cells and particles [94,95]. Specifically, phagocytosis of apoptotic cellular debris induces inflammatory mediators, such as IL1β, IL6, and TNFα, which are released to enhance the activity of phagocytes [94]. Elevated production of pro-inflammatory mediators during the microglia depletion period would likely contribute to an increase in sleep [6,18].

We observed no changes in sleep during weeks 2 or 3 of the depletion period in mice with depleted microglia and nominal changes in sleep during the repopulation period, which strongly indicates that microglia do not play a critical role in regulating sleep under non-inflammatory conditions. Although no study to date has investigated sleep during the depletion of microglia, previous research has demonstrated that microglia do not play a role in the maintenance of diurnal rhythms [96]. In the absence of microglia during PLX treatment, cortical diurnal gene expression remains unchanged compared to mice on the control diet [96]. Similarly, mice treated with PLX had no variations in diurnal and circadian motor activity compared to mice on the control diet [97]. In contrast, diurnal rhythms are disrupted and clock genes and proteins dysregulated in suprachiasmatic nucleus and hippocampus of rats when microglia were depleted using a *Cx3cr1* diphtheria toxin receptor (*Dtr*) transgenic model [98]. The discrepancy between these data and those in the current study may result from the method by which microglia were depleted. Genetic elimination of microglia using a *Cx3cr1 Dtr* transgenic model results in rapid brain pathology, characterized by global ventricular space shrinkage, a pathology not observed following pharmacological depletion of microglia using PLX [99]. Loss of ventricular space as a result of microglia elimination could contribute to disrupted diurnal rhythms. Further, the genetic elimination of microglia in the *Cx3cr1 Dtr* transgenic model is constitutively expressed (i.e., chronic), whereas PLX is acute. Nevertheless, additional studies are warranted to investigate the role of microglia in sleep and circadian rhythms under non-inflammatory conditions.

Our results provide considerable evidence that microglia are necessary for regulating sleep after an inflammatory challenge. Reactive microglia release cytokines that can increase sleep [6,100]; thus, we predicted that microglia depletion would mitigate the inflammatory response and, subsequently, result in less sleep after the administration of LPS. Interestingly, we found that mice with depleted microglia slept significantly more after LPS1 compared to mice on the control diet. Specifically, a robust increase in cumulative sleep occurred during the dark period across all time points in mice with depleted microglia compared to mice with intact microglia. It is well documented that administration of LPS results in inflammation and increases non-rapid-eye-movement (NREM) sleep [40,101]. Microglia depletion may exacerbate the inflammatory response to LPS and worsen inflammation-induced sleep alterations. At 24 h post LPS, IL1β and TNFα are significantly elevated in rats with depleted microglia compared to vehicle controls and rats with intact microglia [102]. Microglia depletion exacerbates the inducible expression of pro-inflammatory cytokines in the brain in response to LPS and sickness behavior assessed by wheel running, which suggest that LPS-induced acute inflammation can develop independently of reactive microglia [102]. IL1β and TNFα are pro-somnogenic and, in the absence of microglia, elevated levels induced by LPS could contribute to the increase in sleep observed in our study.

It is also plausible that in the absence of microglia, anti-inflammatory cytokines do not increase to the same extent in response to LPS administration. In mice, depletion of microglia with PLX did not attenuate the pro-inflammatory response to LPS; however, mice on the PLX diet had a blunted IL10 response [102]. As such, blunted anti-inflammatory responses to LPS may be one mechanism by which microglia elimination affects sleep and inflammation because anti-inflammatory cytokines, such as IL4, IL10, and IL13, attenuate NREM sleep [103,104]. Additional studies are needed to investigate the microglial release profiles of pro- and anti-inflammatory cytokines under inflammatory and non-inflammatory conditions.

It is well documented that a systemic inflammatory challenge activates the immune system and triggers adaptive behavioral changes, referred to as sickness behavior [105,106,107]. These altered behavior adaptations include anorexia, social withdrawal, lethargy, and altered sleep [105]. We observed an increase in sleep after administration of LPS1 in mice on PLX compared to mice on the control diet. We also observed exacerbated sickness behavior in mice on PLX. Two mice on PLX were euthanized following LPS1 because of excessive weight loss and dehydration. It is possible that disruptions to the LPS-induced sleep response in mice on PLX amplified these behavioral deficits. For example, inhibition of an endogenous modulator in the immune system following LPS administration in mice resulted in a more robust change in NREM sleep, as well as an amplified anorectic and hypothermic response [108]. These findings, and our findings, suggest that disruptions in the sleep response to an inflammatory challenge may also result in behavioral deficits and exacerbated sickness behavior.

In the current study, LPS1 resulted in substantial differences in sleep between mice on PLX and mice on the control diet, with profound differences during the dark period. In contrast, after 10 days of microglia repopulation, we administered LPS2 and no differences in sleep were detected during the dark period between mice with repopulated microglia and control mice. We calculated the mean minutes slept during the dark period cumulated across all four days post LPS1 and LPS2. Following LPS1, control mice slept 920 min (95% CI: 861–982), whereas PLX mice slept 1247 min (95% CI: 1165–1335). Cumulatively, during the dark period, mice on PLX slept significantly more (*p* < 0.0001) than control mice. However, following LPS2, mice with repopulated microglia and control mice slept an almost identical amount during the dark period (*p* = 0.88). Control mice slept 787 min (95% CI: 717–864) and mice previously on PLX slept 779 (95% CI: 705–867). These results provide compelling evidence that mice with repopulated microglia have a similar response to control mice following a second inflammatory challenge; i.e., they responded as if naïve to LPS administration.

Limited data exist on the extent to which repeated LPS exposure alters sleep. We further investigated the sleep of control mice after LPS1 compared to sleep after LPS2, specifically focusing on the dark period. We added the cumulative minutes slept in the dark period across all four days and found that control mice slept significantly less after LPS2 compared to LPS1 (*p* = 0.03), suggesting there may have been a blunted immune response following a second exposure to LPS. Microglia can alter their phenotypes depending on prior exposure to an inflammatory stimulus, demonstrating a capacity to develop innate immune memory [109]. Hence, microglia can be desensitized to a stimulus and exhibit immune tolerance or a weaker immune response to subsequent stimuli [110] and demonstrate immune tolerance to LPS; a second LPS dose results in a blunted pro-inflammatory response, both in vitro and in vivo [111]. Our results indicate a second LPS dose changes temporal patterns of sleep but not the daily total minutes slept, findings that also support the hypothesis that there is some level of immune tolerance to repeated LPS.

Following LPS1, mice with depleted microglia had a significant increase in cumulative sleep during the dark period compared to control mice. However, after repopulation, there were no differences in cumulative sleep during the dark period following LPS2 compared to control mice. Mice with repopulated microglia responded to LPS2 similar to controls, with a difference of just 8 min in mean cumulative sleep across all four days of the dark period. Of importance, ~5% of microglia remain in the brain after PLX-induced microglia depletion (Appendix A). We postulate that these remaining microglia were exposed to LPS1 and, when PLX was withdrawn, they repopulated the brain through rapid proliferation and the resulting daughter cells retained innate immune memory.

Altered sleep can induce reactivity in microglia and sleep loss can alter microglial morphology and molecular phenotype [36]. In response to a stimulus, reactive microglia enlarge their cell bodies, retract their processes, and adopt a less complex branching structure [112,113]. We detected changes in microglial morphology following repeated exposure to LPS in mice with repopulated microglia compared to control mice. We quantified the mean number of microglia to confirm that microglia had fully repopulated in the brains of mice previously on the PLX diet. There were no differences in microglia number between PLX and control. However, repopulated microglia had a more reactive phenotype, with significantly less branch endpoints per microglial cell, indicating a less complex morphology compared to controls. These results agree with published findings that, after repopulation, microglia have a more reactive morphology at subacute time points but return to control levels by 28 days [114]. However, studies in mice using computer tracing of microglial branches report no differences in ramification after 14 days of repopulation [115]. The different morphological quantification methods and repopulation periods used in our study, compared to said previous studies, are likely the underlying causes of the conflicting findings. There is substantial evidence that LPS rapidly activates microglia, which persists for the first 72 h post injection [107,116]. In this study, we chose to investigate microglia reactivity at 7 days post LPS2. We predicted that, acutely, microglia from both groups would be reactive, but that by 7 days post-injection, persisting reactivity may indicate a biological difference in the neuroinflammatory response of mice with repopulated microglia compared to control mice. Additional studies are warranted to investigate if the morphological changes we observed between groups endure chronically.

The results of this study should be interpreted in the context of some limitations. Specific sleep stages (i.e., NREM and REM) cannot be distinguished using the non-invasive sleep cages utilized in this study. There is evidence to support that LPS increases NREM sleep while suppressing REM sleep [41]. In our study, the total cumulative sleep could remain unchanged while the ratio of NREM to REM could be altered after LPS. Thus, future studies should include EEG-based determination of altered sleep–wake behavior after LPS administration following microglia depletion. Here, we examined microglia reactivity in three cortical regions; however, morphological changes may have occurred in other brain regions that regulate sleep–wake behavior (e.g., hypothalamus). While PLX is a powerful research tool to target microglia-mediated inflammation, there have been off-target peripheral effects reported that include alterations in a subset of circulating macrophages [117]. It should also be considered that depletions in microglia may affect other cell populations (e.g., astrocytes and neurons). A depletion in microglia does not affect the viability or response of astrocytes [118]; however, PLX administration may increase neural circuit connectivity in mice [119]. Additionally, we used only male mice but evidence for sex differences in response to inflammatory challenges exists [46,120]. Therefore, additional studies are needed to determine if sex is a relevant biological variable for elucidating the responses of mice to LPS under the conditions in our study. Lastly, we did not include a vehicle control for the LPS injections because our comparison of interest was PLX-LPS vs. control-LPS. Additional experiments could be performed to control for stress and handling associated with an injection.

## 5. Conclusions

This study addresses a critical knowledge gap by investigating the functional role of microglia in spontaneous sleep and in LPS-induced altered sleep. We found evidence that microglia do not play a substantial role in the regulation of sleep under non-inflammatory conditions, but strong support that microglia are necessary to regulate sleep after an immune challenge. Repopulating microglia results in LPS-induced changes in sleep that are comparable to those of control mice, suggesting that microglia may play a role in the sleep response to an immune challenge. Consequently, a global depletion in microglia may be an inadvisable therapeutic strategy for treating inflammation-induced sleep disturbances.

## Figures and Tables

**Figure 1 biology-11-01241-f001:**
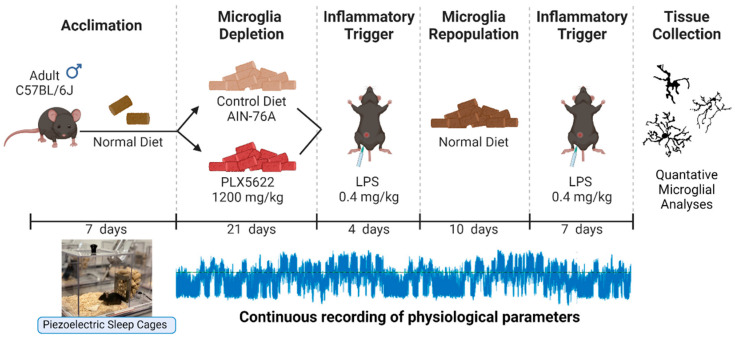
**Study design and experimental timeline**. Adult male mice were acclimated to the piezoelectric sleep cages and fed a normal diet of standard rodent chow for 7 days. Following the acclimation period, mice were randomly assigned to control diet (AIN-76A) or PLX5622 (PLX) diet (1200 mg/kg) for a 21-day microglia depletion period. Mice continued to receive control or PLX diet and were subjected to an inflammatory trigger (lipopolysaccharide; LPS). At 4 days post-LPS injection, all mice were switched back to a normal diet to allow repopulation of microglia in the PLX group. Following a 10-day repopulation period, mice were subjected to a second LPS injection. All mice remained on normal diet until tissue was collected 7 days after the second LPS injection. Brains were processed for immunohistochemistry, microglia were stained with Iba1, and microglia morphology was assessed using quantitative microglial analyses.

**Figure 2 biology-11-01241-f002:**
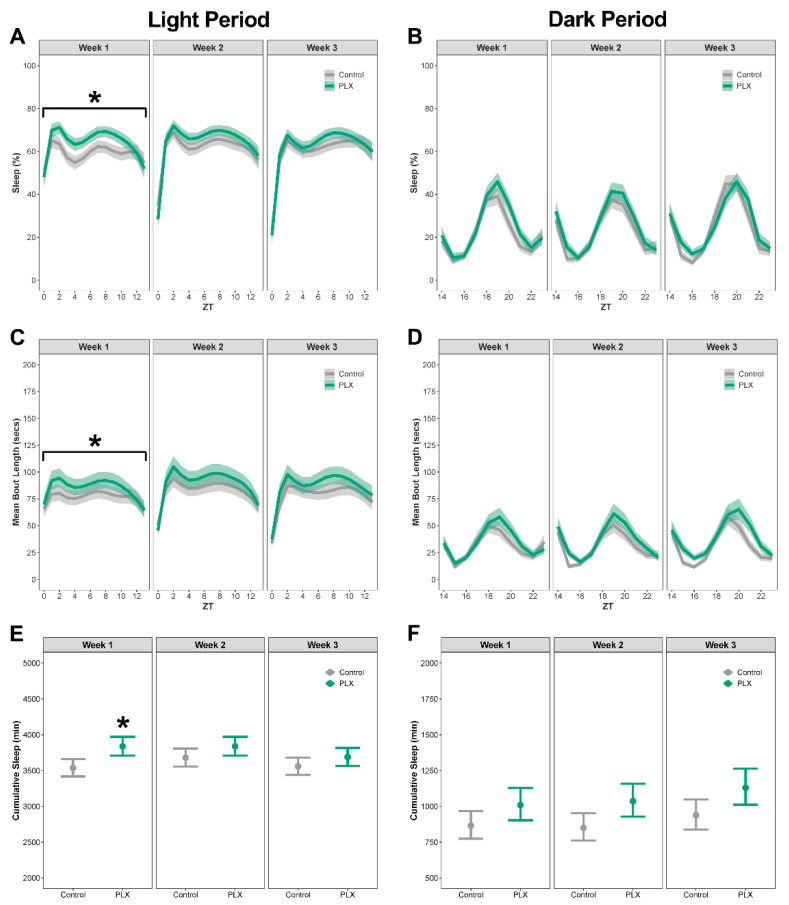
**Microglia depletion resulted in a transient sleep increase during the light period.** Sleep was analyzed by week over the 21-day administration of control or PLX diet. (**A**,**B**) Percent sleep during the light and dark periods. (**C**,**D**) Mean bout length during the light and dark periods. (**E**,**F**) Cumulative minutes slept during the light and dark periods. Results presented as marginal effects point estimates with 95% confidence intervals. * indicates statistically significant differences in means. Control *n* = 15, PLX *n* = 15.

**Figure 3 biology-11-01241-f003:**
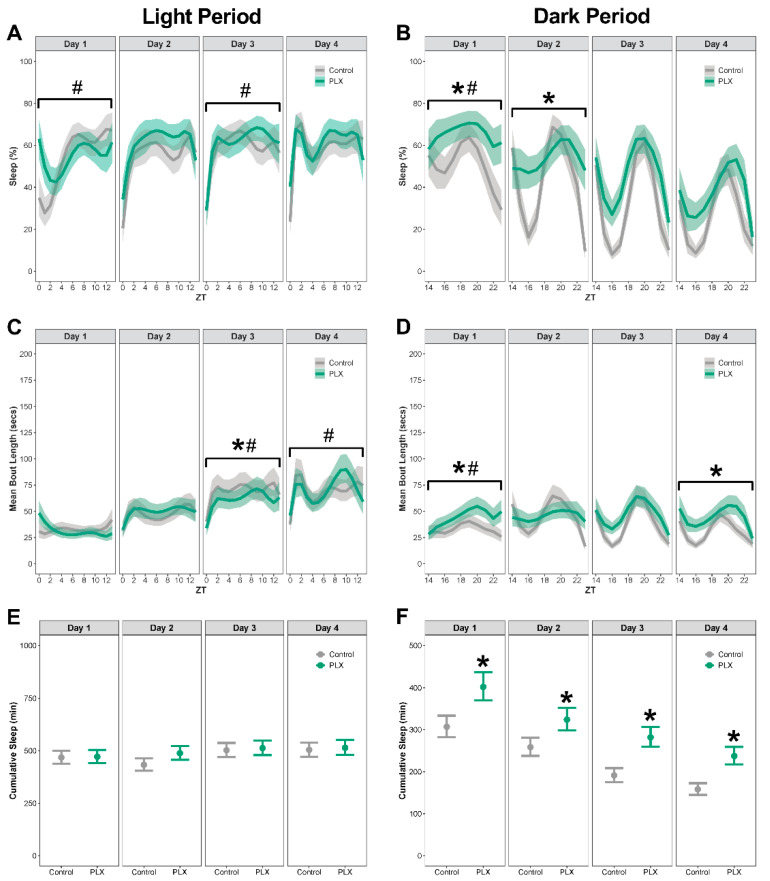
**Mice with depleted microglia slept more during the dark period following LPS1 administration.** Sleep was analyzed by day following intraperitoneal administration of LPS (LPS1). (**A**,**B**) Percent sleep during the light and dark periods. (**C**,**D**) Mean bout length during the light and dark periods. (**E**,**F**) Cumulative minutes slept during the light and dark periods. Results presented as marginal effects point estimates with 95% confidence intervals. * indicates statistically significant differences in means; # indicates statistically significant temporal pattern differences. Control *n* = 15, PLX *n* = 15.

**Figure 4 biology-11-01241-f004:**
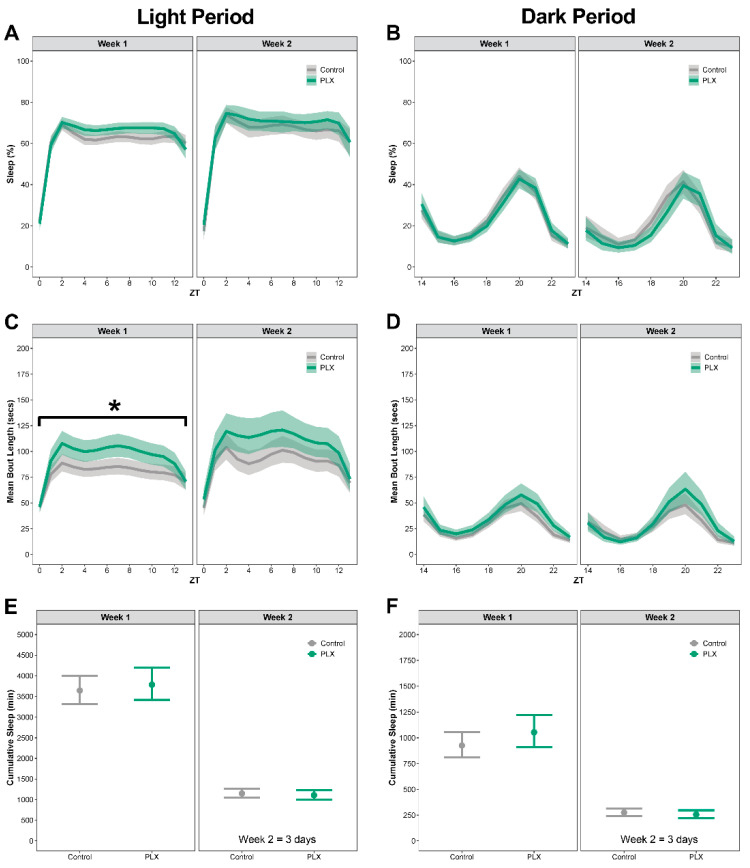
**There were nominal changes in sleep during microglia repopulation.** Mice were placed on normal diet for 10 days to allow microglia to repopulate in mice previously on PLX. Sleep was analyzed by week. Sleep from days 8–10 is denoted as week 2. (**A**,**B**) Percent sleep during the light and dark periods. (**C**,**D**) Mean bout length during the light and dark periods. (**E**,**F**) Cumulative minutes slept during the light and dark periods. Results presented as marginal effects point estimates with 95% confidence intervals. * indicates statistically significant differences in means. Control *n* = 15, PLX *n* = 12.

**Figure 5 biology-11-01241-f005:**
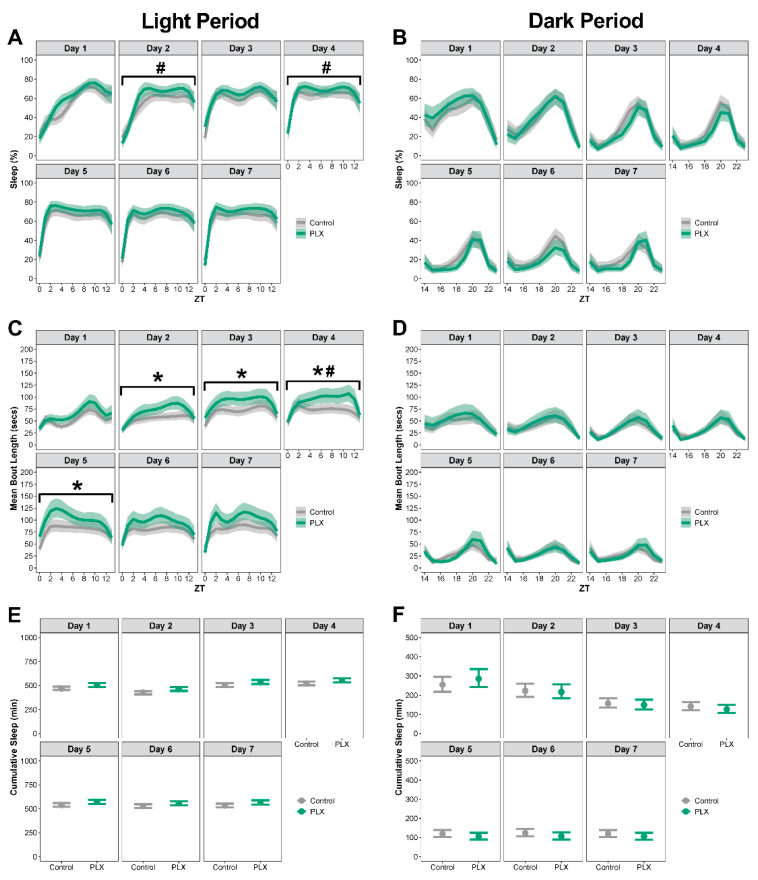
**In mice with repopulated microglia, LPS2 resulted in longer bouts during the light period and no differences in sleep during the dark period compared to controls**. Sleep was analyzed by day following a second intraperitoneal administration of LPS (LPS2). (**A**,**B**) Percent sleep during the light and dark periods. (**C**,**D**) Mean bout length during the light and dark periods. (**E**,**F**) Cumulative minutes slept during the light and dark periods. Results presented as marginal effects point estimates with 95% confidence intervals. * indicates statistically significant differences in means; # indicates statistically significant temporal pattern differences. Control *n* = 14, PLX *n* = 12.

**Figure 6 biology-11-01241-f006:**
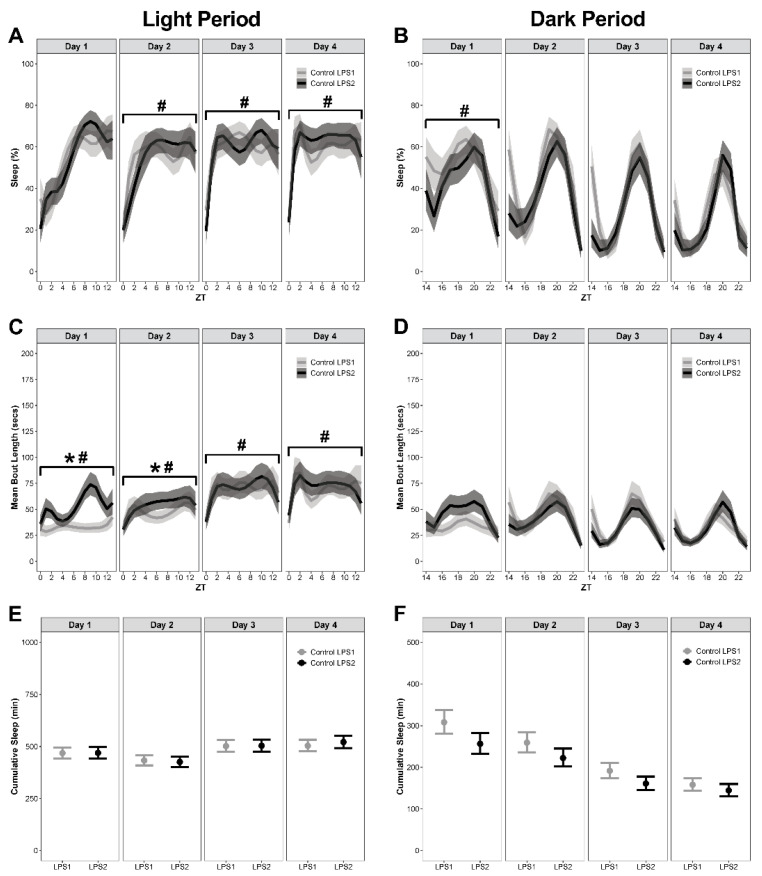
**In control mice, LPS2 did not change the total time spent sleeping but resulted in longer bouts and temporal pattern changes in percent sleep and bout lengths during the light period**. Sleep was analyzed by day and comparisons were made between sleep after LPS1 and LPS2. (**A**,**B**) Percent sleep during the light and dark periods. (**C**,**D**) Mean bout length during the light and dark periods. (**E**,**F**) Cumulative minutes slept during the light and dark periods. Results presented as marginal effects point estimates with 95% confidence intervals. * indicates statistically significant differences in means; # indicates statistically significant temporal pattern differences. Control LPS1 *n* = 15, Control LPS2 *n* = 14.

**Figure 7 biology-11-01241-f007:**
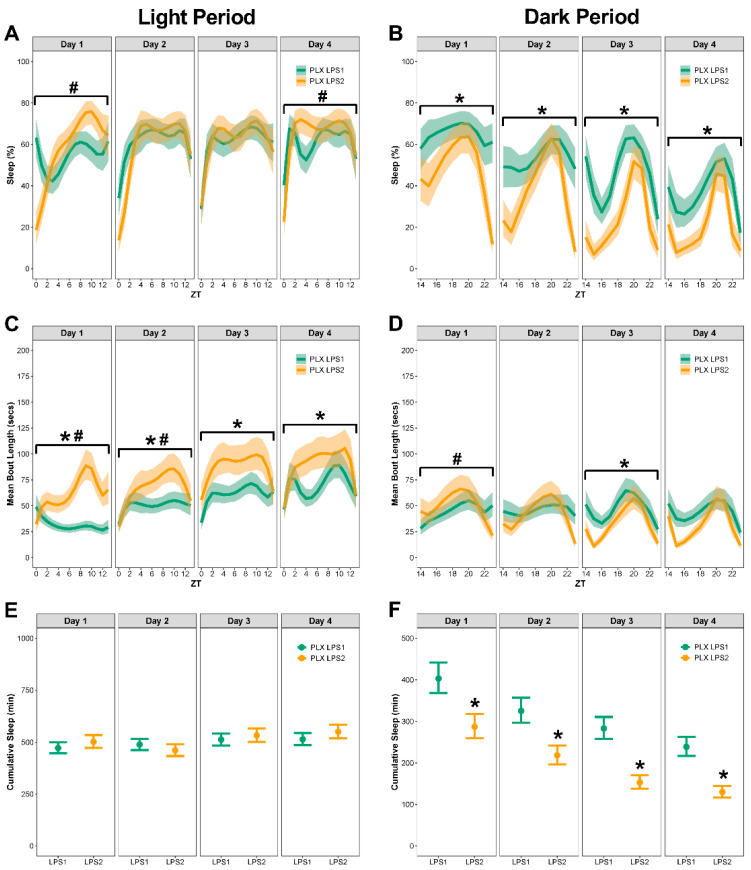
**In mice with repopulated microglia, LPS2 resulted in longer bouts during the light period and less sleep during the dark period**. Sleep was analyzed by day and comparisons were made between sleep after LPS1 and LPS2. (**A**,**B**) Percent sleep during the light and dark periods. (**C**,**D**) Mean bout length during the light and dark periods. (**E**,**F**) Cumulative minutes slept during the light and dark periods. Results presented as marginal effects point estimates with 95% confidence intervals. * indicates statistically significant differences in means; # indicates statistically significant temporal pattern differences. PLX LPS1 *n* = 15, PLX LPS2 *n* = 12.

**Figure 8 biology-11-01241-f008:**
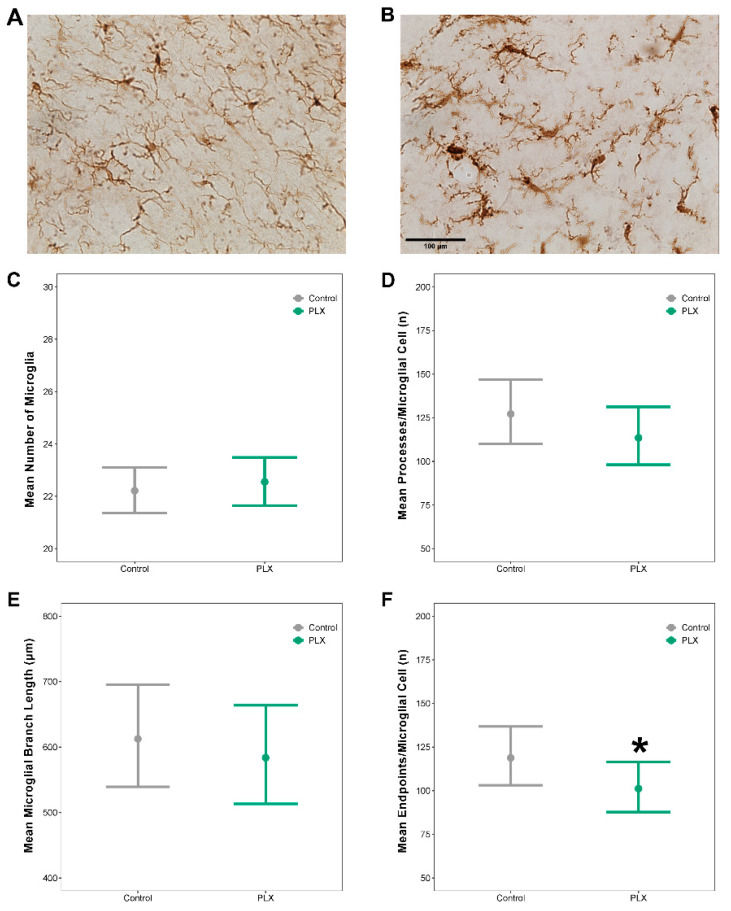
**Repopulated microglia had a reactive morphology with fewer endpoints per microglial cell compared to control mice**. Following repopulation of microglia, tissue was stained with Iba1. Representative Z-stacked images from the cortex of a mouse on (**A**) control diet and (**B**) PLX diet. (**C**) Mean number of microglia per field of view. (**D**) Mean number of processes per microglia, per field of view. (**E**) Mean branch length per microglia, per field of view. (**F**) Mean number of endpoints per microglia, per field of view. Results presented as the marginal effects point estimates with 95% confidence intervals. * indicates statistically significant differences in means. Control *n* = 14, PLX *n* = 11. Scale bar on B = 100 µm.

## Data Availability

Data are publicly available in the Dryad Digital Repository: https://doi.org/10.5061/dryad.tdz08kq2m.

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
