# Peer review of "Microglia Are Necessary to Regulate Sleep after an Immune Challenge"

_biology, 2022, doi:10.3390/biology11081241_

Round 1
Reviewer 1 Report
In this manuscript, Rowe et al study the role of microglia in regulating sleep after an immune challenge with LPS. This is an elegant study that uses PLX to deplete microglia and asks questions about how sleep patterns are changes in their absence and after repopulation.
This is a well-written and well-presented paper that adds significant value to the field of inflammation and sleep. I have only minor comments that need to be addressed.
1) Were a small subgroup of animals euthanized at the end of the PLX feeding to to ensure successful depletion of the microglia?
2) Convert the microglia cell counts in Fig 8 to mm2 instead of per field of view
3) The differences in morphology shown in the micrograph don’t seem to reflect well in the data. Possibly a % area would show better results from the thresholded images.
Author Response
Reviewer 1:
In this manuscript, Rowe et al study the role of microglia in regulating sleep after an immune challenge with LPS. This is an elegant study that uses PLX to deplete microglia and asks questions about how sleep patterns are changes in their absence and after repopulation.
This is a well-written and well-presented paper that adds significant value to the field of inflammation and sleep. I have only minor comments that need to be addressed.
We thank the reviewer for their time spent reviewing our manuscript to strengthen it for consideration in Biology. We have carefully reviewed all comments and have provided responses below.
1) Were a small subgroup of animals euthanized at the end of the PLX feeding to ensure successful depletion of the microglia?
It is well documented that PLX5622 depletes microglia after 7 days of administration, at the dose used in our study (1200 mg/kg). We extended the administration window to 21 days, to ensure >90% depletion. Although many labs have validated this method of microglia depletion, we also confirmed, that in our hands, we achieved microglia depletion after 21 days of administration. We have added a supplementary figure of a representative image of Iba1 staining in the cortex of a mouse following 21 days of PLX administration. This image shows >90% depletion and ensures successful depletion in our PLX-treated group was achieved.
2) Convert the microglia cell counts in Fig 8 to mm2 instead of per field of view
We have published extensively on microglial morphology using the skeletal analysis technique. We have retained the labels on Figure 8 as microglia cell counts per field of view. However, we have updated the Results of the manuscript to also report the microglia cell count per mm2 for the control and PLX groups. Mice with repopulated microglia had 140.85 ± 9.81 microglia cells/mm2 compared to controls that had 139.12 ± 11.78 microglia cells/mm2.
3) The differences in morphology shown in the micrograph don’t seem to reflect well in the data. Possibly a % area would show better results from the thresholded images.
While analyzing the percent coverage or pixel density of Iba1 can show differences in the overall amount of staining, these techniques do not provide any morphological information. For example, a highly ramified microglia with a small cell soma can occupy the same amount of pixels as a classically reactive morphology observed as a larger cell body with shorter branches. For this reason, we chose to focus on skeletal analysis which provides a more quantitative measure of microglial morphology and cell complexity.
Reviewer 2 Report
This manuscript by Rowe et al. studied the effect of microglia on sleeping after inflammatory stimulation. The authors employed PLX5622 to deplete microglia of mice and compare their sleeping with mice on control diet after LPS administration. They found the microglial depletion led to the elongation of sleeping after LPS stimulation. The authors concluded microglia can regulate sleep after the immune challenge. While the finding is interesting, addressing the points below could promote the rigor of the manuscript.
1. According to the literature, https://www.pnas.org/doi/10.1073/pnas.1922788117, the PLX5622 are not only deplete the microglia but also alter other myeloid cells such as macrophages. The author should exclude the possibility that other myeloid cells also affect sleep.
2. The author should prove that PLX5622 didn’t affect neurons and astrocytes during the microglia depletion.
3. The authors studied the microglia during repopulation period after PLX5622 diet and compared them to the microglia with control diet. The authors should provide the image and quantification of microglia in the entire brain. Alternatively, the authors should specify the brain region that they focused on and the rationale for selecting this brain region. They should also provide information of how to randomly select the images of microglia for quantification.
4. Is there any difference between repopulated microglia and microglia on control diet after LPS stimulation? It would improve the understanding of how microglia affect sleep during the second stimulation of LPS.
Author Response
Reviewer 2:
This manuscript by Rowe et al. studied the effect of microglia on sleeping after inflammatory stimulation. The authors employed PLX5622 to deplete microglia of mice and compare their sleeping with mice on control diet after LPS administration. They found the microglial depletion led to the elongation of sleeping after LPS stimulation. The authors concluded microglia can regulate sleep after the immune challenge. While the finding is interesting, addressing the points below could promote the rigor of the manuscript.
We thank the reviewer for their time spent reviewing our manuscript to strengthen it for consideration in Biology. We have carefully reviewed all comments and have provided responses below.
According to the literature, https://www.pnas.org/doi/10.1073/pnas.1922788117, the PLX5622 are not only deplete the microglia but also alter other myeloid cells such as macrophages. The author should exclude the possibility that other myeloid cells also affect sleep. The author should prove that PLX5622 didn’t affect neurons and astrocytes during the microglia depletion.
We agree that it is important to discuss off-target effects of PLX and address how PLX administration may affect astrocytes and neurons during the microglia depletion. These experiments are outside the scope of our current project and have already been conducted by other laboratories. We have added additional points to the Discussion to address the off-target peripheral effects of PLX. We have included current published work that supports that PLX administration does not significantly affect astrocytes during/after microglia depletion. There are mixed findings about neurons and neural circuit connectivity following PLX administration, and we have added this literature to our Discussion.
The authors studied the microglia during repopulation period after PLX5622 diet and compared them to the microglia with control diet. The authors should provide the image and quantification of microglia in the entire brain. Alternatively, the authors should specify the brain region that they focused on and the rationale for selecting this brain region. They should also provide information of how to randomly select the images of microglia for quantification.
We agree that clarification was needed regarding the brain regions analyzed in this study. We have added additional text to the Methods to specify the brain regions that were imaged and analyzed, the methods for selecting images for microglia quantification, and justification and rationale for the brain regions we focused on in this study.
Is there any difference between repopulated microglia and microglia on control diet after LPS stimulation? It would improve the understanding of how microglia affect sleep during the second stimulation of LPS.
In the current study, we did not collect tissue from a subset of mice following the first LPS administration, when microglia were depleted. These studies have already been conducted by another lab and results from these studies are included in our Discussion (https://doi.org/10.1186/s12974-020-01832-2).

Reviewer 3 Report
Rowe and coworkers investigated the roles of microglia in sleep regulation after LPS challenge. They first depleted microglia in the brain using PLX5622 followed by LPS injection to investigate how the inflammatory response in the presence and absence of microglia affects sleep. Subsequently, they switched the diet back to the normal diet to allow to microglia repopulation. That was followed by another LPS trigger. Brains were processed subsequently to examine the microglia morphology and ramifications between the mice fed control and PLX5622 diet. Overall, the authors quantified systematically the sleep parameters during various stages of their experiments and the conclusions were supported.
Major criticisms
1) In Figure 2A, the author noted a higher percentage, longer bouts of sleep in the PLX treated mice when compared to the control (Figure 2A). Given that microglia are effectively eliminated in the PLX mice, can the author reconcile that with the central thesis that microglias are necessary to regulate sleep? Can the author also include some citation, if available, that PLX5622 does not make mice feeling unwell and thus have the tendency to sleep more.
2) In Figure 3A, can the author offer a possible explanation on why there was no difference in sleep percentage during the light period in day 2 when there were some temporal differences in day 1 and 3.
3) In Figure 8, the author showed the morphology of the repopulated microglias in the control and PLX diet mice. It was rather known however that the repopulated microglias after PLX treatment always appear to reactive morphologically when compared to the resident microglias. Given that the authors tried to examine if microglias regulate sleep after challenged with LPS, it would be great if they can examine if the similar difference in mean endpoint/microglia cells is observed in a more relevant brain area such as the suprachiasmatic nucleus which is known to regulate sleep.
4) The word “necessary” in the title and elsewhere in the paper might be too strong given that there was no proper control for LPS treatment (saline or vehicle control). The author should reword the title or add the control experiments.
Minor criticisms
1, When the authors say temporal pattern differences? Is that the difference between two ZT in their study or more generally across all the ZT?
2. Font size in the main figures can be at times too small (Figure 4B)
Author Response
Reviewer 3:
Rowe and coworkers investigated the roles of microglia in sleep regulation after LPS challenge. They first depleted microglia in the brain using PLX5622 followed by LPS injection to investigate how the inflammatory response in the presence and absence of microglia affects sleep. Subsequently, they switched the diet back to the normal diet to allow to microglia repopulation. That was followed by another LPS trigger. Brains were processed subsequently to examine the microglia morphology and ramifications between the mice fed control and PLX5622 diet. Overall, the authors quantified systematically the sleep parameters during various stages of their experiments and the conclusions were supported.
We thank the reviewer for their time spent reviewing our manuscript to strengthen it for consideration in Biology. We have carefully reviewed all comments and have provided responses below.
Major criticisms
1) In Figure 2A, the author noted a higher percentage, longer bouts of sleep in the PLX treated mice when compared to the control (Figure 2A). Given that microglia are effectively eliminated in the PLX mice, can the author reconcile that with the central thesis that microglias are necessary to regulate sleep? Can the author also include some citation, if available, that PLX5622 does not make mice feeling unwell and thus have the tendency to sleep more.
In Figure 2, the changes in sleep are observed during week 1 of PLX administration. We have included a rationale for this finding in our Discussion. Microglia are eliminated during week 1 of the depletion period in our study. The mechanism of eliminating and clearing microglial cells from the brain following PLX-induced depletion is not fully characterized, but there is evidence to support microglia are eliminated as a result of apoptosis. We postulate that during week 1 of the depletion period, microglia were rapidly eliminated and the wide spread cell death and clearance of waste transiently altered sleep. Inflammatory cytokines play a role in phagocytosis and clearance of apoptotic cells and particles. Specifically, phagocytosis of apoptotic cellular debris induces inflammatory mediators, such as IL1β, IL6, and TNFα, which are released to enhance the activity of phagocytes. Elevated production of pro-inflammatory mediators during the microglia depletion period would likely contribute to an increase in sleep.
2) In Figure 3A, can the author offer a possible explanation on why there was no difference in sleep percentage during the light period in day 2 when there were some temporal differences in day 1 and 3.
In our study, the temporal pattern differences for day 2 in the light period failed to reach statistical significance. We agree that in Figure 2A, day two appears to show a difference in percent sleep between groups. While there was not a significant difference between groups, there is still biological difference and likely, sleep is altered in the light period across all three days.
3) In Figure 8, the author showed the morphology of the repopulated microglias in the control and PLX diet mice. It was rather known however that the repopulated microglias after PLX treatment always appear to reactive morphologically when compared to the resident microglias. Given that the authors tried to examine if microglias regulate sleep after challenged with LPS, it would be great if they can examine if the similar difference in mean endpoint/microglia cells is observed in a more relevant brain area such as the suprachiasmatic nucleus which is known to regulate sleep.
We agree that it would be interesting to analyze morphological changes in microglia of the SCN. Due to the size of the SCN in the mouse brain, it would be challenging to quantify microglia morphology using the full field of view skeletal analysis method utilized in our study. We have added additional text to the Methods to specify the brain regions that were imaged and analyzed, the methods for selecting images for microglia quantification, and justification and rationale for the brain regions we focused on in this study.
4) The word “necessary” in the title and elsewhere in the paper might be too strong given that there was no proper control for LPS treatment (saline or vehicle control). The author should reword the title or add the control experiments.
Our manuscript specifically focuses on the role of microglia in sleep after an immune challenge. To investigate sleep after an immune challenge the primary comparisons of importance are between mice with intact microglia administered an immune challenge and mice with depleted/repopulated microglia administered an immune challenge. We have stated this critical comparison in the Methods (Lines 146-148).
Minor criticisms
1, When the authors say temporal pattern differences? Is that the difference between two ZT in their study or more generally across all the ZT?
It is the pattern difference between the two groups across an entire period.
- Font size in the main figures can be at times too small (Figure 4B)
The figure size has been reduced by the journal to meet the formatting requirements. These fonts appear larger on the original figures.

Reviewer 4 Report
Overall, the findings from this study offer a novel insight into the role of microglia in sleep responses in the presence of a systemic LPS challenge. The use of PLX as a tool to deplete microglia in this study further expands on the important use of this tool in studying microglial biology.
Some minor concerns below:
1. Why was 7 days post LPS2 chosen to examine microglial responses? Is there any evidence of more chronic changes in microglial responses in the context of this model?
2. In addition to the cortical regions, did the authors examine microglial responses in other regions, namely the hypothalamus? This may be interesting given the important role of the hypothalamus in sleep regulation.
3. Is there any evidence that disruption in sleep responses are associated with behavioral deficits? If so, could depletion/repopulation alter this?
Round 2
Reviewer 2 Report
More experiments or better control need to be set to prove that neurons, astrocytes and other myeloid cells were not affected by the PLX diet in this research. Otherwise the result can't support the conclusion about role of microglia in the sleeping disorders.
Author Response
/